# Balancing Adaptability and Non-exploitability in Repeated Games

**Anthony DiGiovanni**[1]

**Ambuj Tewari**[1]

[1]Department of Statistics, University of Michigan, Ann Arbor, MI, USA

## Abstract

We study the problem of adaptability in repeated games: simultaneously guaranteeing low regret for several classes of opponents. We add the constraint that our algorithm is non-exploitable, in that the opponent lacks an incentive to use an algorithm against which we cannot achieve rewards exceeding some "fair" value. Our solution is an expert algorithm (LAFF), which searches within a set of sub-algorithms that are optimal for each opponent class, and punishes evidence of exploitation by switching to a policy that enforces a fair solution. With benchmarks that depend on the opponent class, we first show that LAFF has sublinear regret uniformly over these classes. Second, we show that LAFF discourages exploitation, because exploitative opponents have linear regret. To our knowledge, this work is the first to provide guarantees for both regret and non-exploitability in multi-agent learning.

## 1 INTRODUCTION

General-sum repeated games represent interactions between agents aiming to maximize their respective reward functions, with the possibility of compromise over conflicting goals. Despite their simplicity, achieving high rewards in such games is a challenging learning problem due to the complex space of possible opponents. Both the behavior of a given opponent throughout a game, and that opponent's choice of learning algorithm, may depend on one's own algorithm. Crandall [2020] argues, based on empirical studies of repeated game tournaments, that a successful agent must achieve two goals. First, it must optimize its actions with respect to its beliefs about the opponent. Second, it should act such that the opponent forms beliefs motivating a response that is beneficial to the agent.

In particular, multi-agent reinforcement learning (MARL) features the following tradeoff: how to adapt to a variety of potential opponents, while also actively shaping other agents' models of oneself such that they respond with co-operation, rather than exploitation. If an agent commits to a fixed policy to "lead" the other player's best response [Littman and Stone, 2001], it may perform arbitrarily poorly against players that do not converge to such a response. This motivates the design of adaptive algorithms that try to lead, but can retreat to a "Follower" (best response) approach if doing so gives greater rewards [Powers and Shoham, 2005, Chakraborty and Stone, 2010]. An effective algorithm in this class is S++ [Crandall, 2014], which, due to its Follower sub-algorithm, has the drawback that it is exploitable—that is, it rewards agents insisting on unfair bargains ("bully" strategies) [Crandall et al., 2018, Stastny et al., 2021].

A simple motivating example of Follower exploitability is the game of Chicken (Figure 1), between players Row and Column. Suppose Column knows Row will take the apparently optimal action 1 if Column repeats action 2. Column will then want to use the Leader strategy of committing to action 2 to gain the highest reward. Row thus only gets reward 0.25, and if Column has truly committed, an attempt by Row to dissuade this strategy by taking action 2 would give both players reward 0. A cooperative outcome, e.g., alternating between the off-diagonal cells, could be achieved if Row's learning algorithm were designed to *publicly disincentivize* commitments to the exploitative Leader strategy.

| | |
|---|---|
| 0.5, 0.5 | 0.25, 1 |
| 1, 0.25 | 0, 0 |

Figure 1: Reward bimatrix for Chicken.

MARL research has largely neglected the latter half of the adaptability vs. non-exploitability tradeoff. Existing algorithms are either evaluated solely by their rewards *conditional* on given opponents [Powers and Shoham, 2005, Crandall, 2014], or, when the evaluation criterion does account for the incentives of algorithm selection, the pool of

*Accepted for the 38th Conference on Uncertainty in Artificial Intelligence* (UAI 2022).

competitor algorithms typically excludes bully strategies [Crandall and Goodrich, 2010]. Previous MARL algorithms addressing the adaptability half of the tradeoff lack finite-time guarantees on rewards. We aim to provide a theoretically grounded algorithm for repeated games that is both adaptable, by using Leader and Follower sub-algorithms, and non-exploitable. More broadly, this paper addresses a challenge of interest in several areas of machine learning: designing algorithms that account for how the distribution of data the algorithms are applied to may change based on the choice of the algorithms themselves.

**Related work**   Previous algorithms for repeated games have combined Leader and Follower modules, aiming for the following guarantees: worst-case safety, best response to players with bounded memory, and convergence in self-play to Pareto efficiency, i.e., an outcome in which no player can do better without the other doing worse [Powers and Shoham, 2004]. Like ours, these algorithms aim for adaptability, but they do not have regret guarantees — the desired properties are only shown to hold asymptotically. Manipulator [Powers and Shoham, 2005] achieves these properties by starting with a fixed strategy that maximizes the user's rewards conditional on the opponent using a best response, and switching to reinforcement learning (RL) with a safety override if that strategy does not yield its target rewards. Related to the self-play guarantee, we prove a more general property of Pareto efficiency against effective RL algorithms (see Section 2.1). Like Manipulator, our approach tests sub-algorithms sequentially. S++ [Crandall, 2014] has empirically strong performance on the guarantees above. However, neither of these algorithms guarantee non-exploitability.

Although to our knowledge no previous works have proven non-exploitability in our sense, several algorithms are designed to achieve "fair" Pareto efficiency in self-play without using Follower approaches that would be exploitable. Littman and Stone [2005]'s algorithm for computation of Nash equilibria, like our Leader sub-algorithms, enforces a Pareto efficient outcome by punishing deviations. If an agent played this equilibrium, which satisfies properties of symmetry similar to the outcome our Egalitarian Leader sub-algorithm aims for, it would be non-exploitable. However, committing to this equilibrium precludes learning a best response to fixed strategies that offer higher rewards than the cooperative solution, or exploiting adaptive players, which our Conditional Follower and Bully Leader sub-algorithms achieve, respectively. In two-player bandit problems where the reward bimatrix must be learned, UCRG [Tossou et al., 2020] has near-optimal regret in self-play with respect to the egalitarian bargaining solution (Section 2.2). However, it cannot provably cooperate with agents other than itself, learn best responses, or exploit adaptive players.

Our objectives of adaptability and non-exploitability are inspired by work on learning equilibrium [Brafman and Ten-

nenholtz, 2004, Jacq et al., 2020, Clifton and Riché, 2021], a solution concept in which players' *learning algorithms* are in a Nash equilibrium, beyond merely the equilibrium of an individual game itself. This objective accounts for the dependence of the problems faced by multi-agent learning algorithms on the design of such algorithms.

**Contributions**   We propose an algorithm (LAFF) that, to our knowledge, is the first proven to have both strong performance against different classes of players in repeated games and a guarantee of non-exploitability, formalized in Section 2.3. Specifically, these classes consist of stationary algorithms ("Bounded Memory"), unpredictable adversaries ("Adversarial"), and adaptive RL agents ("Follower"). LAFF's modular design allows for extensions to a broader variety of opponent classes in future work. We propose regret metrics appropriate for games against Followers, based on the goal of Pareto efficiency. Our method of proof of adaptability and non-exploitability is novel, applying "optimistic" principles at two levels. First, LAFF starts with the sub-algorithm (or *expert*) that would give the highest expected rewards if the opponent were in that expert's target class ("potential"), then proceeds through experts in descending order of potential. Second, LAFF chooses whether to switch experts by comparing the potential of the active expert with its empirical average reward plus a slack term, which decreases with the time for which the expert is used. For non-exploitability and regret against Followers, we use the properties of an enforceable bargaining solution (see Section 2.2) to upper-bound the other player's rewards.

## 2   PRELIMINARIES

We study a special class of Markov games: repeated games with a bounded memory state representation [Powers and Shoham, 2005] and public randomization.

### 2.1   SETUP AND OPPONENT CLASSIFICATION

Consider a repeated game over $T$ time steps, defined for players $i = 1, 2$ by action spaces $\mathcal{A}^{(i)}$, reward matrices $\mathbf{R}^{(i)}$, and a fixed player memory length $K \in \mathbb{N}$. Here, all $\mathbf{R}^{(i)}(a^{(1)}, a^{(2)}) \in [0, 1]$ are known by both players. At time $t$ the following random variables are drawn: $S_t$ for state, $A_t^{(i)}$ for actions, and $R_t^{(i)} = \mathbf{R}^{(i)}(A_t^{(1)}, A_t^{(2)})$ for rewards. A state space $\mathcal{S} := (\mathcal{A}^{(1)})^K \times (\mathcal{A}^{(2)})^K \times \{0, 1\}^{2K+2}$, and transition probabilities $\mathcal{P}(s'|s, a^{(1)}, a^{(2)})$ between states, are induced by two features: (1) the tuple of both players' last $K$ actions, and (2) the tuple of the last $K$ and current outcome of a randomization signal, for each player. (See Section 2.1.2 of Mailath and Samuelson [2006].) Thus, players condition their actions on their memory of the last $K$ time steps, and a signal that permits correlated action choices.

Formally, let $(w_t^{(1)}, w_t^{(2)}) \in [0, 1]^2$ be weights chosen by the

*Accepted for the 38ᵗʰ Conference on Uncertainty in Artificial Intelligence* (UAI 2022).

respective players at time $t$,[1] and draw $X_t \sim \text{Unif}[0,1]$ independent of all other random variables in the game. Then, letting $y_t^{(i)}$ be the realized value of $Y_t^{(i)} := \mathbb{I}[X_t < w_t^{(i)}]$, the second feature at time $t$ is $(y_{t-K}^{(1)}, ..., y_t^{(1)}, y_{t-K}^{(2)}, ..., y_t^{(2)})$. This allows the players to correlate actions through the public signal $X_t$, even if one player unilaterally generates the signal. For instance, in Chicken (Figure 1), players could flip a fair coin ($w_t^{(1)} = w_t^{(2)} = 0.5$) at each time step and play the pair of actions leading to the top-right cell when it comes up heads, otherwise play the bottom-left cell. In this framework, at each time step each player has a choice of both a weight $w_t^{(i)}$ and policy $\pi_t^{(i)} : \mathcal{S} \to \Delta^{|\mathcal{A}^{(i)}|}$, a mapping from states to distributions over actions.

Given a fixed policy of player 2, a repeated game is a Markov decision process (MDP) given by $(\mathcal{S}, \mathcal{A}^{(1)}, r, p)$ as follows. Let $a^{(i)}(s)$ be the last action of player $i$ that defines state $s$. Here, $r : \mathcal{S} \times \mathcal{A}^{(1)} \to [0,1]$ is $r(s,a) = \mathbf{R}^{(1)}(a^{(1)}(s), a^{(2)}(s))$, and $p : \mathcal{S} \times \mathcal{A}^{(1)} \times \mathcal{S} \to [0,1]$ is $p(s'|s,a) = \sum_{a^{(2)}} \mathcal{P}(s'|s,a,a^{(2)})\pi^{(2)}(a^{(2)}|s)$. A policy is called Markov if it is conditioned only on the current state.

The problem faced by our learner, player 1, depends on which of the following classes player 2's algorithm is in:

1. *Bounded Memory*: (i) Player 2 uses a constant $w^{(2)}$, reported at the start of the game; (ii) $\pi^{(2)}$ is Markov and does not depend on time or player 1's signals $w_t^{(1)}$ or $y_t^{(1)}$; and (iii) for all $s, a^{(2)}$ we have $\pi^{(2)}(a^{(2)}|s) > 0$.[2]

2. *Adversarial*: Player 2 selects actions according to any arbitrary distribution, which may depend on the history of play and on player 1's policy at each time step.

3. *Follower*: A Follower learns a best response when player 1 is "eventually stationary" (formalizing the follower concept in Littman and Stone [2001]), and when the value of that best response meets player 2's standard of fairness. For some fairness threshold $V^{(2)} \geq 0$ (depending on the game), player 2's algorithm has the following properties. Suppose that after time $T_0$, player 1 always plays a Bounded Memory algorithm (without condition 3), which induces an MDP of finite diameter $D$ where player 2's optimal average reward is at least $V^{(2)}$. Then with probability at least $1 - \delta$, player 2's regret up to time $T$ (see Section 2.3) is bounded by $C_1 T_0 + C_2 D(SAT \log(T/\delta))^{1/2}$ for constants $C_1, C_2$.

A repeated game against a Bounded Memory player is equivalent to a communicating MDP [Puterman, 1994]. A

---

[1] We restrict to cases where players commit to a fixed weight, so the effective action space is finite. See the Appendix for details.

[2] This relatively strong condition is needed for a concentration result in our analysis, ruling out cases where players remain in a transient state for an unknown time. We need to know the exit time from the transient states to compute the quantity $\bar{r}_{i,\tau}^{(2)}$ used by one of our experts. Section 5 shows strong results against a Bounded Memory player (FTFT) for which this condition does not hold.

Follower formalizes an agent that models *our* agent as an MDP (Leader), and the regret bound in our definition is of a standard form for RL algorithms [Wei et al., 2020]. Many MARL algorithms take this approach at least partly [Powers and Shoham, 2005, Chakraborty and Stone, 2010, Crandall and Goodrich, 2010], hence this is a reasonable class to consider. For example, Littman and Stone [2005]'s algorithm, which plays a certain sequence of actions and punishes deviations from that sequence, is Bounded Memory — this algorithm does not change its policy in response to the other player, but its policy conditions on past actions. A standard RL algorithm, which would learn the sequence played by Littman and Stone [2005]'s algorithm and converge to an optimal policy against it, and which is a component of more complex repeated games algorithms like Manipulator and S++, is a case of a Follower.

As discussed in Crandall [2020], a large proportion of top-performing algorithms are Bounded Memory (Leaders) or Followers, or switch between the two. These classes illustrate fundamental approaches to multi-agent learning (thus, likely opponents that our algorithm would face): Either an agent behaves consistently, trying to shape the learning opponent's behavior (Bounded Memory), or the agent changes policies in a process of learning how the opponent behaves and computing an optimal response to that opponent, possibly subject to fairness standards as they try to avoid exploitation (Follower). The Adversarial class accounts for opponent behavior between these two extremes, which is difficult to learn in generality, but a worst-case guarantee can still be achieved. We thus restrict to guarantees against formalizations of these classes. Bounds against a wider variety of opponents would be less theoretically tractable, as far as finding the optimal strategy against one class interferes with performance against another. (For example, Powers and Shoham [2005] note that in the repeated Prisoner's Dilemma, it is impossible for an algorithm to guarantee the best response to an opponent that may play either grim trigger — "defect if and only if either player defected last round" — or "always cooperate.") Extending to other opponent classes is an important direction for future work.

## 2.2 BACKGROUND ON BARGAINING THEORY

To define appropriate optimality criteria for these opponent classes and construct corresponding experts, we use several concepts from bargaining theory. We also illustrate these concepts in the game of Chicken from the introduction (Example 2.1). Define the *security values* $\mu_S^{(i)} := \max_{\mathbf{v}_i} \min_{\mathbf{v}_{-i}} \mathbf{v}_1^\mathsf{T} \mathbf{R}^{(i)} \mathbf{v}_2$, i.e., the rewards that each player can guarantee regardless of their opponent's actions, with player 1's maximin strategy as $\mathbf{v}_M^{(1)} = \arg\max_{\mathbf{v}_1} \min_{\mathbf{v}_2} \mathbf{v}_1^\mathsf{T} \mathbf{R}^{(1)} \mathbf{v}_2$. Let $\mathcal{G} := \{(\mathbf{R}^{(1)}(i,j), \mathbf{R}^{(2)}(i,j)) \mid i \in \mathcal{A}^{(1)}, j \in \mathcal{A}^{(2)}\}$, the set of reward pairs achievable by pure actions in the game. An impor-

*Accepted for the 38th Conference on Uncertainty in Artificial Intelligence* (UAI 2022).

tant set of rewards in the computation of enforceable bargaining solutions is the convex polytope $\mathcal{U} := \mathrm{Conv}(\mathcal{G}) \cap \{(u_1, u_2) \mid u_1 \geq \mu_{\mathrm{S}}^{(1)}, u_2 \geq \mu_{\mathrm{S}}^{(2)}\}$, reward pairs that are achievable by randomizing over joint actions and give each player at least their security value. One reward pair satisfying several desirable properties is the egalitarian bargaining solution (EBS) [Tossou et al., 2020], given by $(\mu_{\mathrm{E}}^{(1)}, \mu_{\mathrm{E}}^{(2)}) := \arg\max_{(u_1, u_2) \in \mathcal{U}} \min_{i=1,2}\{u_i - \mu_{\mathrm{S}}^{(i)}\}$.

The reward pairs over which we search for optimal benchmark values, described in Section 2.3, are subject to the following constraint of enforceability. To our knowledge, this definition, including the formalization of enforceability for finite punishment lengths, has not been provided in previous work on non-discounted games. However, see Definition 2.5.1 in Mailath and Samuelson [2006] for the discounted case.

**Definition 1.** *Let $(u_1, u_2) \in \mathcal{U}$ be a convex combination of points in some set of joint actions $\mathcal{X}$. Let $r(\mathcal{X}) := \max_{(x_1, x_2) \in \mathcal{X}}\{\max_{j \neq x_2} \mathbf{R}^{(2)}(x_1, j) - \mathbf{R}^{(2)}(x_1, x_2)\}$ be player 2's deviation profit. Then $(u_1, u_2)$ is $\epsilon$-**enforceable**, relative to a memory length $K$ and $\epsilon > 0$, if:*

$$K u_2 \geq K \mu_{\mathrm{S}}^{(2)} + r(\mathcal{X}) + \epsilon.$$

Intuitively, if player 2 does not deviate from player 1's desired action sequence, player 2 receives $u_2$ on average for each of $K$ steps. If player 2 deviates, gaining at most $r(\mathcal{X})$ profit, player 1 may punish with player 2's security value for $K$ steps. We call the total sequence reward "enforceable" if it exceeds the total deviation reward by at least $\epsilon$. Let $\mathcal{U}(\epsilon)$ be the set of $\epsilon$-enforceable rewards in $\mathcal{U}$. Then, the feasible region $\mathcal{U}(\epsilon)$, used to compute an enforceable version of the EBS, shrinks with increasing $\epsilon$ and decreasing $K$.

The $\epsilon$-enforceable EBS, which we will use to design one of the Leader experts, is found by solving the optimization problem from Section 3.2.4 of Tossou et al. [2020] under the constraint in Definition 1. A similar procedure, applied to the objective of maximizing only player 1's reward, gives the Bully solution for the second Leader expert. We provide details on these solutions in the Appendix.

**Example 2.1.** *In Chicken (Figure 1), both players' security value is 0.25, guaranteed by playing action 1. The EBS is given by 50% weight on the top-right action pair, and 50% on the bottom-left, giving both players 0.625. If player 1 plays its half of either action pair in the EBS, player 2 does worse by deviating (by a margin of at least 0.25), so no punishment is necessary to enforce the EBS. Thus the EBS is enforceable for any $K$ and $\epsilon < 0.375K + 0.25$.*

## 2.3 OBJECTIVES

The metric of regret, which we aim to minimize, varies based on the class of player 2 our algorithm faces. For a player 2 algorithm $\mathfrak{B}$, regret with respect to a benchmark $\mu(\mathfrak{B})$ is $\mathcal{R}(T) := T\mu(\mathfrak{B}) - \sum_{t=1}^{T} R_t^{(1)}$.

**Bounded Memory** By condition 3 for Bounded Memory, player 2 induces a communicating MDP. Let $\Pi$ be the set of time-independent deterministic Markov policies. Then the state-independent optimal average reward is $\mu_*^{(1)} := \max_{\pi^{(1)} \in \Pi} \lim_{t \to \infty} \frac{1}{t} \mathbb{E}_{\pi^{(1)}}(\sum_{i=0}^{t} R_i^{(1)} | S_0)$. Here, $\mu(\mathfrak{B}) = \mu_*^{(1)}$.

**Adversarial** Against an Adversarial player, an appropriate benchmark is the greatest expected value that player 1 can guarantee, no matter player 2's actions. This is player 1's security value: $\mu(\mathfrak{B}) = \mu_{\mathrm{S}}^{(1)}$. Note the distinction from *external regret* used in adversarial bandits and MDPs. While the problem is trivial if player 2 is known to be Adversarial, since one can always play the maximin strategy, our challenge is to maintain low Adversarial regret without losing guarantees on other regret measures. This corresponds to *safety* in multi-agent learning [Powers and Shoham, 2004].

**Follower** The concept of regret against a Follower is more complex. Player 2's sequence of policies can vary significantly based on player 1's actions. Evaluating our algorithm by the maximum average reward in hindsight would have to account for this counterfactual dependence [Crandall, 2014]. However, by considering enforceability, we can define benchmarks by lower bounds on this maximum, constrained by the Follower's fairness value $V^{(2)}$. We consider two cases depending on $V^{(2)}$, focusing for simplicity on the extremes where the Follower either accepts nothing less than the EBS or accepts any enforceable bargain. In principle, our framework could be extended for other $V^{(2)}$ values.

First, the EBS is Pareto efficient, meaning we cannot achieve greater than $\mu_{\mathrm{E}}^{(1)}$ without player 2 receiving less than $\mu_{\mathrm{E}}^{(2)}$. When the EBS can be enforced with a fixed policy, $\mu_{\mathrm{E}}^{(1)}$ is thus an appropriate benchmark if the fairness threshold $V^{(2)}$ is player 2's part of the EBS pair. The EBS is not always enforceable for finite $K$, however. In this case, the enforceable version of the EBS is the maximizer $(\mu_{\mathrm{E},\epsilon}^{(1)}, \mu_{\mathrm{E},\epsilon}^{(2)})$ of the objective $f(u_1, u_2) = \min_{i=1,2}\{u_i - \mu_{\mathrm{S}}^{(i)}\}$ in $\mathcal{U}(\epsilon)$ for some $\epsilon > 0$. For this first case, we therefore consider $V^{(2)} = \mu_{\mathrm{E},\epsilon}^{(2)}$, where player 2 follows conditionally. If $\mathcal{U}(\epsilon)$ is empty, $(\mu_{\mathrm{E},\epsilon}^{(1)}, \mu_{\mathrm{E},\epsilon}^{(2)}) := (\mu_{\mathrm{S}}^{(1)}, \mu_{\mathrm{S}}^{(2)})$. We set $\mu(\mathfrak{B}) = \mu_{\mathrm{E},\epsilon}^{(1)}$.

The second case is $V^{(2)} = 0$, i.e., player 2 follows unconditionally. Here, we compute the maximizer over $\mathcal{U}(\epsilon)$ of $f(u_1, u_2) = u_1$. Let $(\mu_{\mathrm{B},\epsilon}^{(1)}, \mu_{\mathrm{B},\epsilon}^{(2)})$ be the solution to this optimization problem (the *Bully values*), or $(\mu_{\mathrm{B},\epsilon}^{(1)}, \mu_{\mathrm{B},\epsilon}^{(2)}) := (\mu_{\mathrm{S}}^{(1)}, \mu_{\mathrm{S}}^{(2)})$ if no solution exists. We define $\mu(\mathfrak{B}) = \mu_{\mathrm{B},\epsilon}^{(1)}$.

While these regret metrics provide standards for adaptability, we must also formalize non-exploitability. We seek a

*Accepted for the 38$^{th}$ Conference on Uncertainty in Artificial Intelligence* (UAI 2022).

guarantee on an algorithm's performance against its best response. It is unclear how to characterize the best response to an algorithm capable of adapting to several opponent classes. Given this, we focus on a tractable and practically relevant subproblem: guaranteeing that the best response to our algorithm is not a "bully" in the sense discussed in the introduction, which is the most common exploitative strategy in MARL literature [Powers and Shoham, 2005, Littman and Stone, 2001, Press and Dyson, 2012, Littman and Stone, 2005]. Even this weaker guarantee is absent from previous work, and we show numerically in Section 5 that this suffices for our algorithm to be in learning equilibrium with itself (see Section 1) in a pool of top-performing algorithms.

**Definition 2.** *Let player 2 be Bounded Memory, and $\mu_{\mathrm{M}}^{(1)}$ and $\mu_{\mathrm{M}}^{(2)}$ be the expected rewards for players 1 and 2 when player 1 uses $\mathbf{v}_{\mathrm{M}}^{(1)}$ and player 2 uses $\pi^{(2)}$. An algorithm $\mathfrak{A}$ is $(V^{(1)}, \eta_e)$-**non-exploitable** if, whenever $\mu_*^{(1)} < V^{(1)} - \eta_e$ and $\mu_{\mathrm{M}}^{(2)} > \mu_{\mathrm{E},\epsilon}^{(2)}$, for all $c > 0$ player 2's regret with respect to $\mu_{\mathrm{E},\epsilon}^{(2)} + c$ against $\mathfrak{A}$ is $\Omega(T)$.*

Our algorithm is exploitable if player 2 can profit (do better than $\mu_{\mathrm{E},\epsilon}^{(2)}$) from a policy against which we cannot achieve close to some value corresponding to a standard of fairness. The hyperparameter $V^{(1)}$ tunes the tradeoff between exploitability and flexibility to various opponents. Player 2 does *not* profit from exploitation if they incur linear regret.

**Example 2.2.** *In Chicken (Figure 1), let $V^{(1)} = 0.625$ (i.e., the EBS), and consider the following strategies: a) always play action 2, b) always play the opponent's last action, and c) play the best response to the empirical distribution of the opponent's past actions. Strategy (a) is exploitative Bounded Memory. Thus, we argue that an effective algorithm should avoid playing the "best response" of action 1, instead discouraging the use of this strategy by, e.g., consistently playing the EBS (see Egalitarian Leader in the next section). Strategy (b) is also Bounded Memory, but not exploitative since one can achieve at least $V^{(1)}$ against this player on average. Our algorithm should therefore learn the best response to (b). Strategy (c) is a Follower with $V^{(2)} = 0$, thus our algorithm should converge to consistently playing action 2 against (c), achieving the Bully value.*

# 3 LEAD AND FOLLOW FAIRLY (LAFF)

We apply an expert algorithm to a set of experts designed for our target classes. Expert algorithms use an active expert to choose an action at a given time, and switch active experts based on their relative performance [Crandall, 2014]. LAFF switches experts sequentially, going to the next expert in a predefined sequence only if the rewards obtained by its active expert fall short of the current target value. Some of the experts are also designed to guarantee non-exploitability.

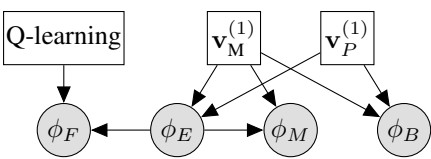

Figure 2: Algorithmic components (white) of LAFF's experts (gray). An arrow from one node to another means the former is used in computation of the output by the latter.

## 3.1 DESCRIPTION OF EXPERTS

LAFF uses an active expert for an epoch of length $H$ before checking whether to switch. Let $\tau$ be the time elapsed since LAFF started using the current instance of the active expert (at time $t_i + 1$), and define $\bar{r}_{i,\tau}^{(1)} := \frac{1}{\tau} \sum_{t=t_i+1}^{t_i+\tau} R_t^{(1)}$ and $\bar{r}_{i,\tau}^{(2)} := \frac{1}{\tau-K} \sum_{t=t_i+K+1}^{t_i+\tau} R_t^{(2)}$. See Figure 2 for a summary of algorithmic elements that these experts depend on.

**Conditional Follower** $(\phi_F)$ Recall the benchmarks $\mu_{\mathrm{B},\epsilon}^{(1)}$, $\mu_{\mathrm{E},\epsilon}^{(1)}$, and $\mu_*^{(1)}$ from Section 2.3. To handle cases where $\mu_*^{(1)}$ against a Bounded Memory player 2 lies between these values, LAFF uses $\phi_F$ multiple times in the sequence (called "instances"). This expert starts off equivalent to Optimistic Q-learning [Wei et al., 2020], whose regret bound (in an MDP with $S$ states and $A$ actions) with probability at least $1 - \delta$ is $\mathcal{R}_Q(\tau, \delta) = \mathcal{O}((SA \log(\frac{\tau}{\delta}))^{1/3} \tau^{2/3})$. After each *subepoch* of length $H^{1/2}$, if $\bar{r}_{i,\tau}^{(1)} < V^{(1)} - \frac{\mathcal{R}_Q(\tau, \delta/T)}{\tau}$, this expert switches to the Egalitarian Leader $\phi_E$ (below) for as long as *any* instance of $\phi_F$ is used. Otherwise, it uses Optimistic Q-learning for the next subepoch.

**Conditional Maximin** $(\phi_M)$ Initially, $\phi_M$ uses the policy $\pi^{(1)}(\cdot|s) = \mathbf{v}_{\mathrm{M}}^{(1)}$ for all $s$. Let $\eta_m > 0$ be a slack variable, chosen based on the class of Adversarial players considered in Theorem 1. After each subepoch, if $\bar{r}_{i,\tau}^{(2)} > \mu_{\mathrm{E},\epsilon}^{(2)} - \eta_m + \sqrt{\frac{\log(T/\delta)}{2(\tau-K)}}$, this expert switches to $\phi_E$ for the rest of the game. Otherwise, it uses $\mathbf{v}_{\mathrm{M}}^{(1)}$ for the next subepoch.

**Egalitarian Leader** $(\phi_E)$ If there is no enforceable EBS, let $\phi_E \equiv \mathbf{v}_{\mathrm{M}}^{(1)}$. Otherwise, let the EBS action pairs be denoted $(a_{\mathrm{E}}^{(1)}(y), a_{\mathrm{E}}^{(2)}(y))$ for $y = 0, 1$, and the weight on the first action pair be $\alpha_{\mathrm{E}}$. While $\epsilon$-enforceability requires that a punishment of length $K$ is sufficient to make a reward pair player 2's best response, this length may not be *necessary*. We therefore consider the least harsh punishment (if any) needed to enforce the EBS, that is, the value $K' \leq K$ satisfying $K' = \max \left\{ 0, \left\lceil \frac{r(\{(a_{\mathrm{E}}^{(1)}(0), a_{\mathrm{E}}^{(2)}(0)), (a_{\mathrm{E}}^{(1)}(1), a_{\mathrm{E}}^{(2)}(1))\}) + \epsilon}{\mu_{\mathrm{E},\epsilon}^{(2)} - \mu_{\mathrm{S}}^{(2)}} \right\rceil \right\}$.

Let $\mathbf{v}_P^{(1)} := \arg\min_{\mathbf{v}_1} \max_{\mathbf{v}_2} \mathbf{v}_1^{\mathsf{T}} \mathbf{R}^{(2)} \mathbf{v}_2$, player 1's punishment strategy. Recall that policies in our framework are

*Accepted for the 38$^{th}$ Conference on Uncertainty in Artificial Intelligence* (UAI 2022).

conditioned on binary signals $Y_t^{(i)}$, whose distributions are determined by players' reported weights $w_t^{(i)}$. Then, for the first $K'$ time steps, with the realized value $y_t^{(1)}$ of the signal given by $w_t^{(1)} = \alpha_E$ for all $t$, $\phi_E$ plays $a_E^{(1)}(y_t^{(1)})$. (This ensures that, if LAFF switches to $\phi_E$ mid-game, player 2 is not punished for having played actions other than the EBS before LAFF started signaling enforcement of the EBS.) Afterwards, $\phi_E$ uses the following stationary policy. If, for any of the past $K'$ timesteps, player 2 has played $A_t^{(2)} \neq a_E^{(2)}(y_t^{(2)})$ — i.e., deviated from the EBS — the distribution over actions for that state is $\mathbf{v}_P^{(1)}$. Otherwise, $a_E^{(1)}(y_t^{(1)})$ is played.

**Bully Leader ($\phi_B$)** This expert is defined like $\phi_E$, but using the Bully solution from Section 2.2 (maximizing the selfish objective). If there is no enforceable solution, given by $(a_B^{(1)}(y), a_B^{(2)}(y))$ for $y = 0, 1$ and $\alpha_B$, let $\phi_B \equiv \mathbf{v}_M^{(1)}$. Otherwise, define $\phi_B$ just as $\phi_E$ for this solution.

### 3.2 ALGORITHM

We design the selection of experts by LAFF (Algorithm 1) such that, for any of our target classes, LAFF eventually commits to the optimal expert against player 2 in a sequence $\{\phi_j\}_j$. Over an epoch, the active expert is executed, and we update this expert's average rewards since it was made active (line 5). Afterwards, LAFF switches to the next expert in the schedule if and only if it rejects the hypothesis that the current expert's expected value exceeds its corresponding target $\mu_j$ (line 7). The false positive rate of this hypothesis test is controlled by a function $\mathcal{B}$, which decreases with $\sqrt{\tau}$. We define $\mathcal{B}$ in the proof of Lemma 1 (see Appendix). Because $\mu_{B,\epsilon}^{(1)} \geq \mu_{E,\epsilon}^{(1)} \geq \mu_S^{(1)}$, and the optimal reward $\mu_*^{(1)}$ against a Bounded Memory player may be greater than $\mu_{B,\epsilon}^{(1)}$ or in between these values, $\{\phi_j\}_j$ prioritizes the order of experts based on the optimal average reward they could achieve against the corresponding player 2 class (line 1).

## 4 ANALYSIS

We will now show that LAFF meets our key criteria of adaptability and non-exploitability. See Appendix for proofs of lemmas and the detailed proof of Theorem 1. Lemma 1 shows that with high probability player 2's rewards against $\phi_E$ are not much greater than the EBS (thus non-exploitability is feasible), and player 1's rewards against a Follower are near the target when the correct Leader is used.

**Lemma 1.** *(Reward Bounds When LAFF Leads)* If player 1 uses $\phi_E$ over a sequence of length $\tau + K'$ starting at time $t^* + 1$, then with probability at least $1 - \frac{3\delta}{T}$:

$$\sum_{t=t^*+K'+1}^{t^*+K'+\tau} R_t^{(2)} \leq K' + 1 + \tau \mu_{E,\epsilon}^{(2)} + 3\sqrt{\tfrac{1}{2}\tau \log(\tfrac{T}{\delta})}.$$

---

**Algorithm 1** Lead and Follow Fairly (LAFF)

1: **Init** target schedule $\{\mu_j\}_j = \{\mu_{B,\epsilon}^{(1)}, \mu_{B,\epsilon}^{(1)}, \mu_{E,\epsilon}^{(1)}, \mu_{E,\epsilon}^{(1)}, \mu_S^{(1)}\}$, expert schedule $\{\phi_j\}_j = \{\phi_F, \phi_B, \phi_F, \phi_E, \phi_F, \phi_M\}$, expert index $j = 1$, $\tau = 0$, $R_\tau = 0$
2: **for** $i = 1, 2, \ldots, \lceil T/H \rceil$ **do**
3:     **for** $t = (i-1)H + 1, \ldots, \min\{iH, T\}$ **do**
4:         Run expert $\phi_j$
5:         $R_\tau \leftarrow R_\tau + \mathbf{R}^{(1)}(A_t^{(1)}, A_t^{(2)})$
6:     $\tau \leftarrow \tau + H$
7:     **if** $j < |\{\phi_j\}_j|$ and $\frac{R_\tau}{\tau} < \mu_j - \mathcal{B}(\tau)$ **then**
8:         $j \leftarrow j + 1$, $\tau \leftarrow 0$, $R_\tau \leftarrow 0$

---

*If player 2 is a Follower with $V^{(2)} = 0$, and player 1 uses $\phi_B$, then with probability at least $1 - \frac{5\delta}{T}$, we have $\bar{r}_{i,\tau}^{(1)} \geq \mu_{B,\epsilon}^{(1)} - \mathcal{B}(\tau)$. If $V^{(2)} = \mu_{E,\epsilon}^{(2)}$, and player 1 uses $\phi_E$, then with probability at least $1 - \frac{5\delta}{T}$, we have $\bar{r}_{i,\tau}^{(1)} \geq \mu_{E,\epsilon}^{(1)} - \mathcal{B}(\tau)$.*

Lemma 2 guarantees that with high probability, LAFF follows or uses the maximin strategy against non-exploitative players, and punishes exploitative players.

**Lemma 2.** *(False Positive and Negative Control of Exploitation Test)* *Consider a sequence of $k$ epochs each of length $H$. Let $m_F^*$ or $m_M^*$ be, respectively, the index of the subepoch within this sequence at the start of which $\phi_F$ or $\phi_M$ switches to punishing with $\phi_E$, if at all (if not, let $m_F^*$ or $m_M^* = \infty$). Let $\eta_e \geq \frac{2\mathcal{R}_Q(H/2, \delta/T)}{H} + \sqrt{\frac{2S^2 A \log(c_0/\delta)}{c_1 H}}$, where $c_0, c_1$ are defined as in Theorem 5.1 of Mannor and Tsitsiklis [2005], and $\eta_m \geq \sqrt{\frac{\log(T/\delta)}{2(H/2-K)}} + \sqrt{\frac{64e \log(N_q/\delta^2)}{(1-\lambda)(H/2-K)}}$, where $\lambda$ and $N_q$ are constants with respect to time defined in Lemma 4 (see Appendix).*

*Then, suppose player 2 is Bounded Memory, and $\phi_F$ is used. If $\mu_*^{(1)} < V^{(1)} - \eta_e$, then with probability at least $1 - \delta$, $m_F^* \leq \lceil \frac{H^{1/2}}{2} \rceil$. If $\mu_*^{(1)} \geq V^{(1)}$, then with probability at most $\frac{kH^{1/2}\delta}{T}$, $m_F^* < \infty$. If $\phi_M$ is used, and $\mu_M^{(2)} > \mu_{E,\epsilon}^{(2)}$, then with probability at least $1 - \delta$, $m_M^* \leq \lceil \frac{H^{1/2}}{2} \rceil$.*

*Suppose player 2 is Adversarial, with a sequence of action distributions $\{\pi_t^{(2)}\}$ such that, for any $M \geq H^{1/2} - K$ and $i$, $\frac{1}{M}\sum_{t=i+1}^{i+M} \mathbf{v}_M^{(1)\mathsf{T}} \mathbf{R}^{(2)} \pi_t^{(2)} \leq \mu_{E,\epsilon}^{(2)} - \eta_m$. Then, if $\phi_M$ is used, with probability at most $\frac{kH^{1/2}\delta}{T}$, $m_M^* < \infty$.*

Our main result, Theorem 1, claims that 1) against each of our target classes, LAFF achieves a regret bound of the same order as Optimistic Q-learning in single-agent MDPs [Wei et al., 2020], and 2) LAFF satisfies non-exploitability.

**Theorem 1.** *Let $\mathcal{C}$ be the set of player 2 algorithms that are any of the following:*

*Accepted for the 38th Conference on Uncertainty in Artificial Intelligence* (UAI 2022).

- *Adversarial, with a sequence of action distributions $\{\pi_t^{(2)}\}$ such that $\frac{1}{M}\sum_{t=i+1}^{i+M} \mathbf{v}_{\mathrm{M}}^{(1)\mathsf{T}} \mathbf{R}^{(2)} \pi_t^{(2)} \leq \mu_{\mathrm{E},\epsilon}^{(2)} - \eta_m$ for any $M \geq T^{1/4}$ and $i$,*

- *Follower, with $V^{(2)} \in \{0, \mu_{\mathrm{E},\epsilon}^{(2)}\}$, or*

- *Bounded Memory, with $\mu_*^{(1)} \geq V^{(1)}$.*

*Let $\eta_m$ and $\eta_e$ satisfy the conditions of Lemma 2. Then, with probability at least $1 - 5\delta$, LAFF satisfies:*

$$\max_{\mathcal{C}} \mathcal{R}(T) = \mathcal{O}(\mathcal{R}_Q(T, \delta/T)).$$

*Further, with probability at least $1 - 6\delta$, LAFF is $(V^{(1)}, \eta_e)$-non-exploitable when there exists an enforceable EBS.*

If there is no enforceable EBS, $\mu_{\mathrm{E},\epsilon}^{(2)} = \mu_{\mathrm{S}}^{(2)}$ and so we cannot guarantee player 2 does worse than $\mu_{\mathrm{E},\epsilon}^{(2)}$ in expectation. The class of Adversarial players for which Theorem 1 holds is technically restrictive. However, non-exploitability requires that for each strategy (expert) used by our algorithm that could be exploited, including Conditional Maximin, we exclude from our target class some subset of opponents. That is, we cannot guarantee low Adversarial regret against players who receive more than the EBS value against maximin, because such players may exploit us.

*Proof Sketch.* For each opponent class, we need to show that with high probability LAFF does not lock in to a suboptimal expert for that class. If LAFF locks in to an expert for which the corresponding target value $\mu_j$ is *greater* than the opponent's benchmark $\mu(\mathfrak{B})$, this implies LAFF consistently receives rewards such that "regret" with respect to $\mu_j$ grows like $\mathcal{R}_Q$, by design of $\mathcal{B}(\tau)$. But since the benchmark is less than $\mu_j$, the true regret is also bounded as desired.

We therefore only need to consider the cases of $\mu_j \leq \mu(\mathfrak{B})$. First, we know that each expert achieves at most $\mathcal{R}_Q$ regret against its target opponent class, by, respectively: the definitions of $\mathcal{R}_Q$ (for non-exploitative Bounded Memory) and maximin (for Adversarial), and Lemma 1 (for Followers). Lemma 2 ensures with high probability that $\phi_F$ and $\phi_M$ do not switch to $\phi_E$ when not exploited, so they inherit the desired regret bounds.

Then, we need only show that once LAFF reaches the expert whose target class matches the opponent (thus guaranteeing low regret using that expert), with high probability LAFF does not switch. But if using the corresponding expert gives LAFF low regret with respect to $\mu(\mathfrak{B}) \geq \mu_j$, then its rewards are sufficiently high that the condition for switching experts (line 7 of Algorithm 1) never holds. The first claim of the theorem follows.

To show non-exploitability, suppose LAFF locks in to the first instance of $\phi_F$. By Lemma 2, $\phi_F$ detects evidence of exploitation sufficiently early that the remaining time left

in the game is linear in $T$. After detecting exploitation, $\phi_F$ plays the same policy as $\phi_E$. But by Lemma 1, against this policy player 2 cannot guarantee an average reward greater than $\mu_{\mathrm{E},\epsilon}^{(2)}$ plus a term that vanishes at a rate $T^{1/2}$. The second claim of the theorem follows for the other possible locked-in experts as well by considering two facts. First, whenever $\phi_E$ or $\phi_B$ is used, Lemma 1 again bounds player 2's rewards, since by Pareto efficiency of the EBS player 2's rewards from the Bully solution cannot exceed $\mu_{\mathrm{E},\epsilon}^{(2)}$. Second, if LAFF reaches $\phi_M$, again Lemma 2 ensures sufficiently fast detection of exploitation with high probability. $\square$

# 5 NUMERICAL EXPERIMENTS

Code for the experiments in this section is available on Github.[3] We evaluate LAFF by three empirical metrics. First, we find LAFF's empirical regret against one algorithm from each target class. Second, LAFF and a set of top-performing repeated games algorithms compete in a round-robin tournament. For each algorithm, we find its rewards against its best response algorithm in this set, and check if it is in a learning equilibrium by applying a Nash equilibrium solver [Knight and Campbell, 2018] to the matrices of empirical rewards for algorithm pairs. These criteria evaluate exploitability: more exploitable algorithms have lower rewards against algorithms that optimize against them, and an exploitable algorithm cannot be in equilibrium with itself unless the fairness threshold $V^{(1)}$ is low. Finally, we perform a replicator dynamic simulation [Crandall et al., 2018]. Each generation, the algorithms' fitness values are computed as averages of the round-robin scores weighted by the distribution of the population of algorithms. Then, the population distribution is updated in proportion to fitness. This evaluates how well a given algorithm performs when the distribution of its opponents is determined by those algorithms' own performance. Exploitability is thus implicitly penalized by accounting for opponents' incentives. Details on the implementation of these experiments are in the Appendix. We set $V^{(1)} = \mu_{\mathrm{E},\epsilon}^{(1)}$.

Our set of competitors to LAFF consists of Bounded Memory (Bully, Forgiving Generalized Tit-for-Tat or FTFT), Follower (M-Qubed, Q-Learning, Fictitious Play), and expert (Manipulator, S++) algorithms. See Appendix for details and sources. We chose these algorithms because, first, they performed well in a repeated games tournament [Crandall et al., 2018], and second, they cover our opponent classes. S++ and Manipulator do not fall cleanly into any of those classes, but they are the closest comparisons in previous literature to LAFF, since they adapt to a variety of opponents by switching between Leader and Follower experts.

To ensure sufficient diversity of test games, we choose games based on the taxonomy of Figure 1 in Bruns [2010]. Six game families are categorized by the structures of their

---

[3] https://github.com/digiovannia/ad_expl

*Accepted for the 38th Conference on Uncertainty in Artificial Intelligence* (UAI 2022).

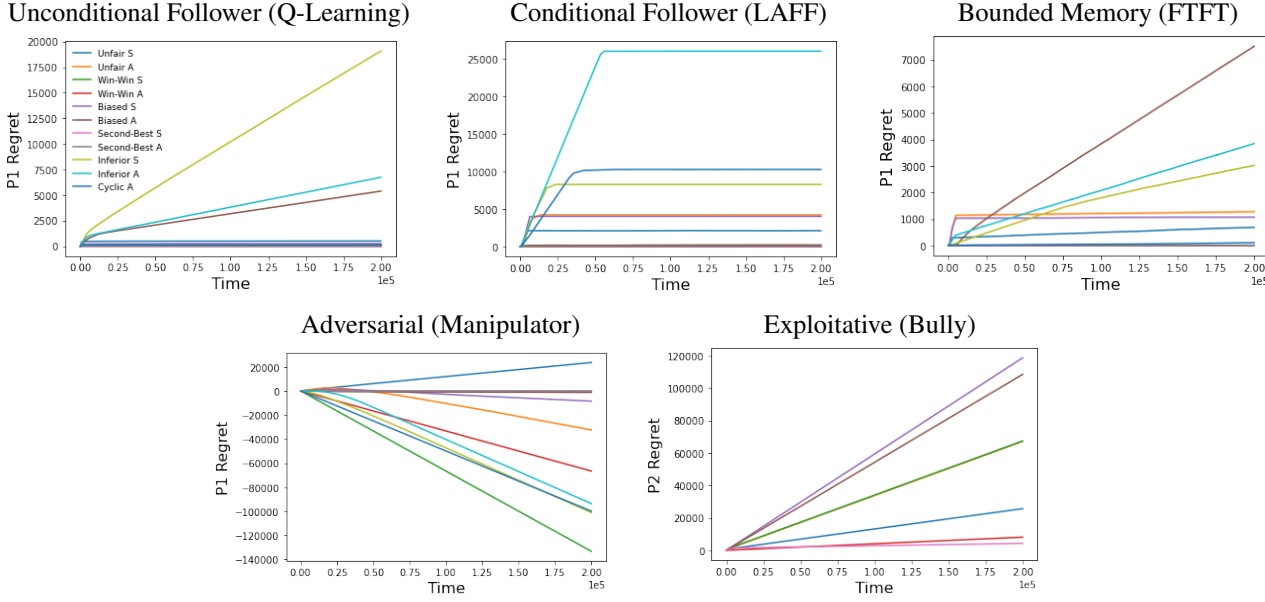

Figure 3: The first four plots show LAFF's average regret, in each of 11 games detailed in the Appendix, for the following opponents: Unconditional Follower (Q-Learning), Conditional Follower (LAFF), Bounded Memory (FTFT), Adversarial (Manipulator). The last plot shows the regret of an Exploitative (Bully) algorithm against LAFF.

Nash equilibria. We use two games from each family, one with symmetric rewards and one with asymmetric, except Cyclic, which has no symmetric games (see Appendix).

**Regret Bounds** Figure 3 shows LAFF's regret, averaged over 50 trials, in games against an algorithm from each target class, and the regret of an exploitative Bounded Memory algorithm against LAFF. We chose Manipulator as "Adversarial" because it does not play the EBS and is not a pure Leader or Follower. However, in the symmetric Unfair game, the empirical rewards indicate that Manipulator attempts to exploit LAFF, so LAFF punishes Manipulator at the expense of the Adversarial regret guarantee. From the plot evaluating player 2's regret, we also exclude four games where player 2's Bully solution equals the EBS, since in these cases $\mu_*^{(1)} \geq V^{(1)}$ (player 1 is not exploited by playing the optimal policy). In most games, LAFF's regret eventually plateaus, while the exploitative player has linear regret, showing that LAFF is non-exploitable. In three games, LAFF has linear regret against an Unconditional Follower and non-exploitative Bounded Memory player. This may be due to the practical difficulty of choosing hyperparameters for tests used to decide when to switch to the next expert; these tests depend on some unknown quantities, so for our experiments, we tuned $\mathcal{B}(\tau)$ on a training set of four games that are not included in the set of 11 games for these results (see Appendix). Longer time horizons may be required for the conditions on $\eta_e$ in Lemma 2 to hold. We used a horizon of $T = 2 \cdot 10^5$ to be on the same approximate scale as experiments in other works on repeated games [Crandall and Goodrich, 2010, Littman and Stone, 2005, Crandall, 2014].

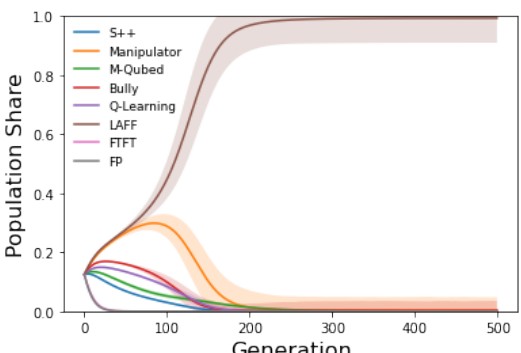

Figure 4: Replicator dynamic results, where the bold curves are average population shares and shaded regions are plus and minus one standard deviation.

**Round Robin** Table 1 shows the average rewards of each algorithm pair across the 11 games and 50 trials, which provide an empirical bimatrix for the *learning game*, i.e., a meta-game in which users choose algorithms to deploy across different repeated games. An algorithm's reward against its best response (highlighted in blue) measures how much it bullies when possible and avoids exploitation. Both as player 1 and player 2, LAFF is second by this metric, behind Bully. We also highlight the pure strategy Nash equilibria of this learning game (in bold), noting that LAFF is in a learning equilibrium with itself. Unfortunately, the pairing in which Q-Learning follows Bully is also an equilibrium. Thus there is an equilibrium selection problem, e.g., both users might choose Bully and receive very low rewards. However, in

*Accepted for the 38$^{th}$ Conference on Uncertainty in Artificial Intelligence* (UAI 2022).

Table 1: Rewards of algorithm pairs, averaged over games and trials (pure learning equilibria in are highlighted in bold text, and each algorithm's reward against its best response is in blue)

| | S++ | Manipulator | M-Qubed | Bully | Q-Learning | LAFF | FTFT | FP |
|---|---|---|---|---|---|---|---|---|
| S++ | 0.75, 0.76 | 0.73, 0.80 | 0.73, 0.81 | 0.65, 0.77 | 0.82, 0.76 | 0.71, 0.8 | 0.70, 0.68 | 0.72, 0.55 |
| Manipulator | 0.87, 0.68 | 0.76, 0.71 | 0.77, 0.65 | 0.65, 0.77 | 0.89, 0.67 | 0.70, 0.65 | 0.71, 0.60 | 0.76, 0.55 |
| M-Qubed | 0.88, 0.68 | 0.68, 0.68 | 0.80, 0.74 | 0.65, 0.80 | 0.79, 0.75 | 0.76, 0.73 | 0.78, 0.65 | 0.62, 0.56 |
| Bully | 0.86, 0.61 | 0.83, 0.60 | 0.85, 0.61 | 0.48, 0.44 | **0.91, 0.63** | 0.61, 0.49 | 0.72, 0.55 | 0.76, 0.56 |
| Q-Learning | 0.82, 0.77 | 0.73, 0.83 | 0.79, 0.67 | **0.68, 0.85** | 0.83, 0.74 | 0.71, 0.84 | 0.81, 0.67 | 0.64, 0.56 |
| LAFF | 0.87, 0.65 | 0.71, 0.66 | 0.74, 0.72 | 0.55, 0.61 | 0.90, 0.66 | **0.77, 0.74** | 0.80, 0.70 | 0.75, 0.57 |
| FTFT | 0.64, 0.70 | 0.49, 0.71 | 0.59, 0.76 | 0.60, 0.71 | 0.59, 0.78 | 0.61, 0.78 | 0.80, 0.75 | 0.46, 0.72 |
| FP | 0.70, 0.73 | 0.66, 0.74 | 0.66, 0.55 | 0.63, 0.73 | 0.69, 0.57 | 0.61, 0.71 | 0.71, 0.60 | 0.68, 0.55 |

practice it may be easier for users to coordinate on both using LAFF, because there is no conflict over choosing which side is the Leader (Bully) versus the Follower (Q-Learning).

**Replicator Dynamic** On average over 1000 runs, LAFF converges to 100% of the population in the pool of algorithms (Figure 4), based on fitness computed as the *minimum* of an algorithm's average reward over the set of games when playing as player 1 versus player 2. This metric matches the motivation for the EBS; algorithm users will not know *a priori* which of the two "sides" of the game they will be in. Thus, they may prefer their algorithm to cooperate with itself (maximize an egalitarian objective), instead of bullying its copy in hopes of being on the side of the bully.

## 6 DISCUSSION

When choosing algorithms for multi-agent interactions, users will have to trade off robustness to the variety of possible algorithms they might face, with avoiding providing other users incentives to exploit them [Stastny et al., 2021]. We have presented an algorithm for repeated games that balances these desiderata. Both properties can facilitate cooperation between learning agents, while still allowing them to accept generous offers. If LAFF faces an agent who "follows" fair, Pareto efficient bargaining proposals, the Egalitarian Leader leads them to a mutual benefit over their security values. If the other agent's fairness standard is different, the Conditional Follower can follow this alternative proposal using RL if it is not exploitative; otherwise, the exploitation penalty encourages the other player to be more cooperative. Against exploitable agents, the Bully Leader can benefit from a more self-interested bargain. Finally, if the other player is unwilling to cooperate at all but is not exploitative, Conditional Maximin ensures safety. In future work, more experts can be added based on agent classes that we have neglected. For example, while LAFF includes Leader experts only for the extreme cases in which player 2 has a high or minimal fairness standard, one could add Leaders for other bargaining solutions.

The biggest limitations of our approach are restrictive assumptions required for our non-exploitability criterion, and the strictness of this criterion. The margin $\eta_e$ is small only for sufficiently large time horizons, hence the linear regret in some of our experiments. Though LAFF successfully punishes players against whom it receives less than fair rewards, this is only strategically necessary when such players *benefit* from playing this way (genuine "exploitation"). It may not be practically necessary to modify the experts to not punish when the opponent also does worse, because an opponent would not have an incentive to lead with a Pareto inefficient policy. Finally, we note that our approach is not intended to provide the optimal balance of the adaptability-exploitability tradeoff; in particular, keeping a fixed fairness threshold may not be ideal if it prevents an algorithm from cooperating with algorithms that follow other intuitively "fair" standards [Stastny et al., 2021].

### Author Contributions

Both authors conceived and carried out the research project jointly. A.D. wrote the paper and code for numerical experiments. A.T. helped edit the paper.

### Acknowledgements

A.D. acknowledges the support of a grant from the Center on Long-Term Risk Fund.

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
