# OpenReview forum: "Balancing Adaptability and Non-exploitability in Repeated Games"
_auai.org/UAI/2022/Conference — UAI 2022 Poster_

### Official Review · Reviewer_SyaD · 2022-04-08

**Q2(1) Originality/Novelty:** 3
**Q2(2) Significance/Impact:** 3
**Q2(3) Correctness/Technical Quality:** 3
**Q2(6) Clarity Of Writing:** 4
**Q6 Overall Score:** 6
**Q8 Confidence In Your Score:** 2

**Q1 Summary And Contributions:**

The paper suggests an approach for adapting with low regret and non-exploitability for several types of opponents.  The paper (self-proclaimed) is the first to provide guarantees for both regret and non-exploitability in multi-agent learning.

**Q2 Assessment Of The Paper:**

More detailed information regarding each of these aspects is given below:

**Q2(4) Quality Of Experiments (Optional):**

3: Good: The experimental evaluation is adequate, and the results convincingly support the main claims.

**Q2(5) Reproducibility:**

3: Good: Key resources (e.g., proofs, code, data) are available and key details (e.g., proofs, experimental setup) are sufficiently well-described for competent researchers to confidently reproduce the main results.

**Q3 Main Strengths:**

- The motivation, background, and methods are presented clearly and at a good level of detail.
- The paper (self-proclaimed) is the first to provide guarantees for both regret and non-exploitability in multi-agent learning.
- Good discussion of the results, implications, and limitations


**Q4 Main Weakness:**

- Related work should be better organized - it is hard to understand what the most relevant body of research is and whether other work has explored low regret (with and without exploitability) in other settings. This should be made more explicit.
- More justification should be provided for the choice of opponent classes and baseline methods.
- Much important information is referred to the appendix. This is understandable given the page limit.


**Q5 Detailed Comments To The Authors:**



**Q7 Justification For Your Score:**

While I'm not an expert on this topic, I think the paper offers enough merit to warrant publication.

**Q9 Complying With Reviewing Instructions:**

1: Yes.

---

### Official Review · Reviewer_Mbfk · 2022-04-11

**Q2(1) Originality/Novelty:** 3
**Q2(2) Significance/Impact:** 2
**Q2(3) Correctness/Technical Quality:** 3
**Q2(6) Clarity Of Writing:** 2
**Q6 Overall Score:** 6
**Q8 Confidence In Your Score:** 3

**Q1 Summary And Contributions:**

The topic is on game theory and multi-agents. For playing a game repeatedly, there is a proposal for an algorithm that incorporates finding a strategy for different kinds of policies faced by another player (with families of policies for other player), while at the same time avoiding exploitation. Exploitation can occur e.g., for leading or following policies The proposed algorithm aims to incorporate non-exploitability. The algorithm is theoretically analyzed and experimentally evaluated.

**Q2 Assessment Of The Paper:**

More detailed information regarding each of these aspects is given below:

**Q2(4) Quality Of Experiments (Optional):**

3: Good: The experimental evaluation is adequate, and the results convincingly support the main claims.

**Q2(5) Reproducibility:**

3: Good: Key resources (e.g., proofs, code, data) are available and key details (e.g., proofs, experimental setup) are sufficiently well-described for competent researchers to confidently reproduce the main results.

**Q3 Main Strengths:**

The two main strengths, in my mind, are that the proposed algorithm is both theoretically analyzed and experimentally evaluated. On both sides, interesting results were shown.

**Q4 Main Weakness:**

This paper does not fall into my core area of research. Nevertheless, I found the paper to not be very accessible; however this might be different for experts in the field.

-There are no examples that exemplify the used definitions. This makes the paper rather difficult to follow. In particular, there are many definitions inline in the text without much explanations.
-I did not find the theoretical results to be interpreted (explained) in much detail. While there is some explanation for Lemma 1 and Lemma 2, it seems not much is said about Theorem 1 (both regarding intuition and impact).

Overall, my impression is that the paper might not be sufficiently accessible for a general (U)AI conference. This also makes assessment of the (formal) results difficult.

**Q5 Detailed Comments To The Authors:**

Overall, I think the problem tackled by the paper is relevant. Adaptation to various strategies of another player (opponent) seems very much worthwhile to investigate. The scientific contributions (algorithm, analysis, experimentation) appear interesting and useful. Several related works are discussed, suggesting a good embedding in the literature.

My recommendation would be to improve presentation (in my mind significantly). While others might disagree, my impression is that the paper is not very accessible. I certainly understand that in a conference version one needs to balance contributions and explanations/descriptions. Nevertheless, I recommend to significantly invest in making the manuscript easier to grasp, e.g., by including examples, and more intuitions on the used definitions.

**Q7 Justification For Your Score:**

Prior to discussion and seeing other reviews, my recommendation is that of "weak accept": it does seem to me that the paper makes a useful contribution to the field, but accessibility is, in my opinion, mediocre. Regarding the significance, from the presented material it is not clear whether the impact is mostly within a specific subfield of AI, or more broad. Overall, I currently arrive at a weak accept.

**Q9 Complying With Reviewing Instructions:**

1: Yes.

---

### Official Review · Reviewer_ArPc · 2022-04-14

**Q2(1) Originality/Novelty:** 4
**Q2(2) Significance/Impact:** 4
**Q2(3) Correctness/Technical Quality:** 3
**Q2(6) Clarity Of Writing:** 4
**Q6 Overall Score:** 8
**Q8 Confidence In Your Score:** 3

**Q1 Summary And Contributions:**

This paper  study the problem of adaptability in repeated games: simultaneously guaranteeing low regret for several classes of opponents, authors have presented an algorithm for repeated games that balances these desiderata. Both properties can facilitate cooperation between learning agents, while still allowing them to accept generous offers. The solution presented is an expert algorithm (LAFF), which searches within a set of sub-algorithms that are optimal for each opponent class.

**Q2 Assessment Of The Paper:**

More detailed information regarding each of these aspects is given below:

**Q2(4) Quality Of Experiments (Optional):**

3: Good: The experimental evaluation is adequate, and the results convincingly support the main claims.

**Q2(5) Reproducibility:**

3: Good: Key resources (e.g., proofs, code, data) are available and key details (e.g., proofs, experimental setup) are sufficiently well-described for competent researchers to confidently reproduce the main results.

**Q3 Main Strengths:**

Authors have provided key resources, such as proofs of lemmas and annexes to help understand the main ideas.

**Q4 Main Weakness:**

1. Authors have stated that the biggest limitations of their approach are restrictive assumptions required for their non-exploitability criterion, and the strictness of this criterion.
2. Authors do not share the code to easily reproduce the results.

**Q5 Detailed Comments To The Authors:**

In general, the manuscript is well written and providing the proofs of lemmas.
Authors could share the code to easily reproduce the results.
In Figure 3, authors could use subcaptions to clearly identify the different opponent class: Unconditional Follower, Conditional Follower (LAFF), Bounded Memory (FTFT), Adversarial (Manipulator) and Exploitative (Bully).

**Q7 Justification For Your Score:**

Authors study a special class of Markov games: repeated games with a bounded memory state representation and public randomization. Authors have presented an algorithm for repeated games and have shown that both properties used can facilitate cooperation between learning agents, while still allowing them to accept generous offers.

**Q9 Complying With Reviewing Instructions:**

1: Yes.

---

### Decision · Program_Chairs · 2022-05-15

**Decision:**

Accept (Poster)

**Comment:**

Meta Review: There is uniform agreement among reviewers that this paper provides solid contributions and should be accepted.